# A Preliminary Study on Acute Otitis Media in Spanish Children with Late Dinner Habits

**DOI:** 10.3390/ijerph191710721

**Published:** 2022-08-28

**Authors:** Ruth Díez, Sergio Verd, Jaume Ponce-Taylor, Antonio Gutiérrez, María Llull, María-Isabel Martin-Delgado, Olga Cadevall, Jan Ramakers

**Affiliations:** 1Department of Pediatrics, Son Espases University Hospital, 07120 Palma de Mallorca, Spain; 2Department of Pediatrics, Quiron Rotger Hospital, 07012 Palma de Mallorca, Spain; 3Pediatric Unit, La Vileta Surgery, Department of Primary Care, 07013 Palma de Mallorca, Spain; 4Balearic Islands Health Research Institute (IdISBa), 07120 Palma de Mallorca, Spain; 5A&E Unit, Department of Primary Care, 07014 Palma de Mallorca, Spain; 6Department of Hematology, Son Espases University Hospital, 07120 Palma de Mallorca, Spain; 7Baleares Medical Council, 07012 Palma de Mallorca, Spain; 8Pediatric Unit, Esporles Surgery, Department of Primary Care, 07190 Mallorca, Spain; 9Pediatric Unit, Santa Ponsa Surgery, Department of Primary Care, 07180 Mallorca, Spain

**Keywords:** otitis media, late dinner, common cold, otitis media, Mediterranean diet, oxidative stress, chronotype, circadian clock, inflammatory disease

## Abstract

The timing of caloric intake plays an important role in the long-term process that leads to communicable diseases. The primary objective of this study was to analyse whether children who ate dinner early were at lower risks of acute respiratory infections than children who ate dinner late during the COVID-19 pandemic. Methods: This cross-sectional study was conducted from July to December 2020 on children attending Majorcan emergency services. Our survey on dinner time habits was carried out by using self-administered questionnaires. Results: A total of 669 children were included in this study. The median dinner time was 8:30 pm. Late dinner eaters accounted for a higher proportion of acute otitis media (7% vs. 3%; *p* = 0.028) than early dinner eaters. Other infectious diseases were not associated with dinner time habits. Conclusions: We make a preliminary estimate of the link between late dinner habits and acute otitis media in children. However, no conclusions about causality can be established due to the observational design of the study, and further research is needed in order to confirm the different issues raised by our initial exploration of an emerging research area.

## 1. Introduction

Almost every aspect of life on earth is regulated by circadian rhythms. In this context, it is becoming clearer that the timing of caloric intake plays an important role in the long-term process that leads to non-communicable diseases [1]. Very recent research reported that higher carbohydrate consumption at breakfast was associated with a significantly lower C-reactive protein (CRP) vs. higher carbohydrate consumption at dinner and that every one-unit increase in percent energy consumed after 5 pm may predict an increase in breast cancer risk associated with low-grade inflammation [2]. In the same vein, Danish researchers have defined low-grade inflammation as a CRP level between 3 and 10 mg/L and have reported that low-grade inflammation was associated with increased risk of hospital-based treatment for infection as well as with the overall use of antimicrobials among men [2,3]. Research efforts around the influence of late dinners on paediatric infectious diseases are ongoing for immune modulation by underlying chronobiologic mechanisms [4].

The evening chronotype has a tendency to wake up later and prefers to time peak activity during the evening, while the morning chronotype usually wakes up earlier and prefers activities earlier in the day. A relevant characteristic of subjects with the evening chronotype is not only the consumption of more calories after 8:00 pm but also lower fruit and vegetable consumption [5,6]. Consequently, the evening chronotype has the lowest adherence to the Mediterranean diet (MD) [7]. The evening chronotype has a tendency to have more cardio-metabolic health problems and to host immune disorders compared to the morning chronotype [8]. On the other hand, abandonment of the MD contributes to the appearance of paediatric common colds and their bacterial complications [9]. Both pathways reflect the anti-inflammatory effects of the MD [10].

Overall, these findings point to a synergy between early dinner and MD adherence, which might contribute to preventing various childhood infectious diseases, particularly respiratory infections. Specifically, both recurrent and severe episodes of acute otitis media (AOM) in childhood are associated with an unbalanced diet, with a higher preference for high-fat products and subsequently with overweight [11,12]. Moreover, there is a bidirectional relationship between AOM and childhood overweight. It seems that middle ear inflammation may involve damage to the chorda tympani nerve with taste sensation impairment [13]. Alterations in taste sensation may explain why otitis media patients need a considerably higher fat intake than healthy patients [14].

During the COVID-19 pandemic, it has been shown that children who went to bed earlier had fewer psychosocial problems [15]. In addition to this, we propose to investigate whether children who were early dinner eaters (EDEs) were at lower risks of infectious disorders when compared to children who were late dinner eaters (LDEs) during the COVID-19 pandemic.

Finally, the secondary objective of this research was to assess Majorcan children’s dinner time variations during the early stages of the COVID-19 pandemic.

## 2. Materials and Methods

Study strategy: This is a secondary analysis of a cross-sectional study that tests the hypothesis that the risk of COVID-19 is attenuated for ever breastfed children [16] and that has also reported a link between late dinner and prehypertension among Majorcan children [17]. This study was conducted from July to December 2020 on children attending Majorcan emergency services of nine healthcare centres for community Paediatrics, a Private Hospital, a District Hospital and two University Hospitals, where no-cost COVID-19 tests were widely available. The extent of illness in children attending emergency services ranged from mild (ambulatory care) to moderate (admission to the paediatric ward) and severe (admission to intensive care). Eligible patients were those younger than 18 years of age who presented to a participating emergency room and who were tested for SARS-CoV-2 because of COVID-19 symptoms, routine pooled testing or scheduled hospital procedures on the 2nd and 16th of each month from July to December 2020. Table 1 represents the data about how many patients were eligible and recruited on these dates.

Data collection: The decline in primary care consultations at the beginning of the COVID-19 pandemic has been widely reported. It is not clear what drove this reduction. We do not know whether families were worried about accessing services or whether access was restricted because of capacity. As it could not be otherwise, at that point, most paediatric patient care across our community was delivered in the accident and emergency departments. In this respect, a common pitfall of emergency room studies is that the base population may not compose a sample representative of the patient population of interest. Studies that fail to use a random sample of available medical records may lack internal validity, and external validity will be compromised if records were taken from a setting with an atypical practice style. Additionally, research reports in emergency medicine settings are less likely to be blinded or longitudinal, or to have been funded and have fewer subjects than studies performed outside the frame of emergency medicine. However, the availability of the data at the onset of the coronavirus emergency led us to accept that a cross-sectional study in the emergency room was an appropriate method of data collection to answer the proposed research question.

Health interview: On enrolment, team members undertook in-person interviews to collect clinical data.

A list of all components of the Balearic Health Service medical interview [18] (not validated) was checked by the clinician (self-introducing; checking the patient’s name and date of birth; obtaining interview consent; asking for family and social history, history of present illness, past medical history and about allergies; asking whether they had any questions).

The data collected included especially past medical history of chronic diseases, current weight and height, timing, symptoms, laboratory tests and imaging studies of current illness. Any diagnosis made was validated by trained medical staff by means of a detailed clinical interview and physical examination. Although cross-sectional studies are useful in identifying associations that can then be more rigorously studied using a cohort study, the authors recognise that carrying out the study in the emergency room introduces an important bias with respect to causality. Personal and social confounding factors that may influence both the potential cause and effect were examined: age, gender, infant feeding, body mass index (BMI), public vs. private healthcare and household composition. The classification of late and early dinner eaters was carried out by using self-administered questionnaires (a paper-and-pencil version), following most of the prior studies of dinner timing that have also been completed using questionnaires. Children were classified into two groups according to the median dinner time 8:30 p.m. Children who had dinner before or at 8:30 p.m. were considered EDEs, while those who had dinner after 8:30 p.m. were considered LDEs. In addition, we calculated the following variables: (A) mean dinner time; (B) dinner phase deviation, the standard deviation of the mean dinner time; and (C) measures of skewness and kurtosis.

Sample size: At the time of surveying, the wide variation in paediatric COVID-19 incidence (1–16%) [18,19] precluded a robust sample size estimate. Hence, we have opted for a convenience sample of children screened for COVID-19.

Data analysis: Statistical analyses were performed with IBM SPSS v.28 statistical software. Categorical data are presented as percentages. Proportions were compared using Fisher’s exact or chi-squared tests. Quantitative and qualitative variables were compared by the Mann–Whitney U test. A *p* value < 0.05 (two-tailed) was considered significant for all statistical analyses.

Ethics: The study was approved by the Balearic Conjoint Health Research Ethics Review Board (COVID IB4221/20PI) and carried out in compliance with the ethical standards of the Declaration of Helsinki. There is no direct benefit of study participation. A research team member at each institution contacted the guardian/caregiver/child in person to obtain written informed consent and assent, as appropriate.

Pre-registration: AsPredicted Trials Registry number of COVID IB4221/20PI is #62721.

## 3. Results

A total of 669 children of ages from 8 days to 17 years old were included in the study. Table 2 shows the general characteristics of the study subjects. The mean age was 54 months, and 44% of the participants were girls. The timing of dinner is summarized in Figure 1; the most frequent dinner time was 8:00 p.m. (26%), followed by 9:00 p.m. (25%) and 8:30 p.m. (21%). The median of dinner time was 8:30 p.m. (IQR = 1); the mean of dinner time was 8.49 (SD: 0.73); and measures of skewness and kurtosis were −0.114 and 0.571, respectively.

Sociodemographic characteristics according to dinner time: Compared with LDE (after 8:30 p.m.), children whose families reported early dinner habits (before or at 8:30 p.m.) were more likely to attend private healthcare facilities and to be slightly younger. In addition, LDEs were found to live in households with fewer children under five years old, as opposed to EDEs. No statistically significant differences were found in gender distribution, type of infant feeding, or children’s BMI across dinner times.

Association between dinner time and clinical characteristics: LDEs accounted for a higher proportion of acute otitis media (AOM) (*p* = 0.028) than EDEs did. The remaining clinical characteristics were not associated with dinner time habits (Table 2 and Table 3).

## 4. Discussion

Temporal distribution of dinner: In our sample, there was a very sharp peak at 8:00 p.m.–9:00 p.m., when around 72% of respondents were found to eat dinner. This finding is in agreement with previous reports of pronounced peak times of dinner in Mediterranean countries compared to Central/Northern European countries, where dinner times were more evenly spread across the evening [20]. Studies on meal timing in Spanish children are scarce; to the best of our knowledge, only Martinez-Lozano et al. [21] have recently reported that the mean dinner time for 397 school-aged children (8–12 years) was 21:07 (95%CI: 20:08; 22:06). There is, therefore, no major difference between their results for dinner time and our own results; the median dinner eating time in our study was roughly half an hour earlier (20:30 to ~21:00) than in the aforementioned study.

The impact of dinner time on paediatric respiratory infections: We did not find increasing rates of flu-like syndrome or lower respiratory tract infection among LDE. However, a new finding is that a late dinner habit was associated with an increased risk of AOM (7% vs. 3%; *p* = 0.028). Overall, there were 33 (4.9%) physician-diagnosed episodes of AOM among 669 paediatric emergency visits between July and December 2020. This figure compares with findings from comprehensive research on emergency visits in which AOM diagnoses encompassed 6.8% of all paediatric emergency room visits with significant variation in the month of presentation (peak: January; trough: September) [22]. In addition, the incidence of AOM peaks between ages 6 and 12 months, and overcrowding is a risk factor for otitis [23]; remarkably, there was no difference in the number of bedrooms between study groups within our sample, but EDEs were a little younger than LDEs.

Allergic rhinitis [24] and oxidative stress [25] are key factors in the pathogenesis of otitis that might provide an understanding of how a late dinner contributes towards developing otitis. Regarding allergic rhinitis, Wasilewska et al. [26] have recently studied the times of meals consumed by children with allergy symptoms, and they have found incorrect dietary habits such as eating < 1 h before bedtime among most children with respiratory allergies; researchers suggested that their findings indicated that eating immediately before bed might have contributed to gastroesophageal reflux, resulting in further local respiratory inflammation and vasomotor changes in this group of children. With regard to late dinners as a risk factor for oxidative stress, a 2004 study [27] of the systemic antioxidant status of patients with otitis showed that children requiring tube insertion have significantly increased erythrocyte levels of malondialdehyde, one of the most frequently used indicators of oxidative stress, compared to healthy controls. Similarly, a 2013 article [28] reported that serum malondialdehyde levels were significantly higher in patients with chronic otitis media than in healthy controls, and a 2019 study has found that patients with otitis have significantly increased serum levels of antioxidant enzymes involved in the defence against oxidative stress [25]. In this same field of research, it has been proven that childhood obesity and total cholesterol levels are significantly higher in children with AOM [29]. Accordingly, our study shows a trend towards higher BMI (*p* = 0.07) among LDE. The main mechanisms for developing otitis media in patients with obesity include gastroesophageal reflux and/or alteration in cytokine profile [30]. The importance of circadian rhythm in the pathophysiology of diseases of the airways that are subject to systemic inflammation, such as chronic obstructive pulmonary disease, chronic sinusitis, idiopathic pulmonary fibrosis or asthma has been reported many times [31]. The inclusion of otitis in this select group, therefore, should be given serious consideration.

Limitations: Since missing data can be thought of as a form of selection bias, we acknowledge that our large amounts of missing data may result in incorrect conclusions. However, the degree to which selection bias can compromise validity is determined by the context and varies with each individual variable. Hence, there is no way to set an acceptable proportion of missing data [31]. Spain is located relatively westward within its time zone, resulting in sunset occurring at a later time as compared to many other countries within the same time zone. Spain also belongs to the Central European Time zone, which runs along the border between Germany and Poland and does not adequately reflect the solar time on Spain’s longitude. The late meal timing in Spain is thus less extreme than what the clock time suggests as compared to solar time. Since we used questionnaires to analyse dinner timing, the dinner-time patterns identified in the present study should not be viewed as indisputable. We looked at primary complaints in paediatric emergency visits; thus, this study may underestimate underlying diagnoses in this cohort of children [32]. Another limitation of this study is the case definition for AOM—clinical judgement made by physicians might not have been accurate, but primary care is a good proxy to general population epidemiology; when specialised clinics are included, the better accuracy of the diagnosis takes a toll on the representativeness of the studied population. Although the relationship between dinnertime habits and AOM in children is supported by our findings, no conclusions about causality can be established due to the observational design of the study. In addition, considering that a number of factors were correlated with each other and that we did not correct for multiple comparisons, further research is needed in order to confirm the different issues raised by this preliminary study.

## 5. Conclusions

Our study shows that a late dinner is a risk factor for AOM in children visiting the emergency room. However, this is an initial exploration and more research should be carried out to substantiate its findings.

If the evidence presented in this report were to be confirmed, child health would benefit from consuming a greater proportion of calories earlier in the day, as compared to consuming a large number of calories later at night. However, compliance with this recommendation may not be feasible for many families and represents a paradigm shift from traditional eating patterns in many parts of the world.

## Figures and Tables

**Figure 1 ijerph-19-10721-f001:**
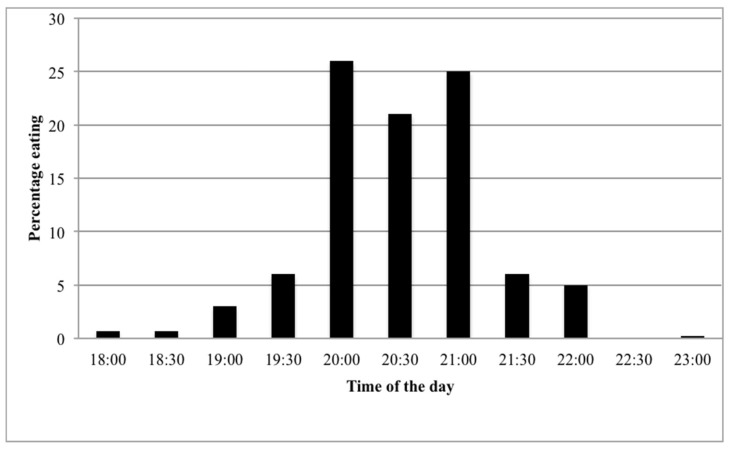
Dinner time.

**Table 1 ijerph-19-10721-t001:** Participant recruitment.

Recruitment Date (Year 2020)	Number of Eligible Participants	Number (%) of Recruited Participants
2nd August	103	72 (69.9)
16th August	117	73 (62.3)
2nd September	84	43 (51.1)
16th September	159	87 (54.7)
2nd October	168	101 (60.1)
16th October	189	58 (30.6)
2nd November	167	89 (53.2)
16th November	176	70 (39.7)
2nd December	182	57 (31.3)
16th December	45	19 (42.2)
Total	1390	669 (48.1)

**Table 2 ijerph-19-10721-t002:** Baseline characteristics of total, early dinner eaters and late dinner eaters.

Variables	Total	Early Dinner (*n* = 409)	Late Dinner (*n* = 260)	*p*
Ever breastfed children	451 (68%)	272 (67%)	179 (70%)	0.49
Age, months	54 (95.5)	39 (110.7)	44 (82.5)	0.011 *
BMI (kg/m^2^)	17.2 (3.9)	16.8 (3.8)	17.6 (4.4)	0.07
Healthcare:				<0.001 ***
Public Private	454 (68%) 215 (32%)	252 (55%) 157 (73%)	202 (44%) 58 (27%)	
Bedrooms in the household:				0.71
0–3 >3	592 (89%) 75 (11%)	363 (89%) 44 (11%)	229 (88%) 31 (12%)	
Children under 5 y in the household:				<0.001 ***
0	206 (35%)	111 (30%)	95 (45%)	
1	300 (51%)	198 (53%)	102 (48%)	
2	70 (12%)	56 (15%)	14 (7%)	
3	8 (1%)	8 (2%)	0 (0%)	

Data are presented in numbers (%), or median (interquartile range). Abbreviations: BMI, body mass index; y, years of age; * *p* < 0.05; *** *p* < 0.001.

**Table 3 ijerph-19-10721-t003:** Differences in clinical characteristics between early and late dinner eaters.

	Early Dinner, before or at 8:30 p.m. *n* = 409	Late Dinner, after 8:30 p.m. *n* = 260	*p*
Comorbidity	84 (20%)	49 (19%)	0.62
Positive PCR test for SARS-Cov-2	7 (2%)	7 (3%)	0.42
Respiratory rate/minute	30 (16)	26 (19.2)	0.056
% oxygen saturation	98 (1)	98 (0)	0.51
Asthenia	39 (9%)	29 (11%)	0.51
Headache	34 (8%)	24 (9%)	0.67
Myalgia	11 (3%)	8 (3%)	0.81
Sore throat	55 (13%)	43 (16%)	0.31
Otitis	14 (3%)	19 (7%)	0.028 *
Breath shortness	28 (7%)	14 (5%)	0.51
Abdominal pain	43 (10%)	31 (12%)	0.61
Diarrhoea	38 (9%)	22 (8%)	0.78
Pain score	5 (4)	5 (3)	0.23
Disease severity			0.074
mild	382 (93%)	253 (97%)	
moderate	26 (6%)	7 (3%)	
severe	1 (0.2%)	0 (0%)	

Data are presented in numbers (%) or median (interquartile range). Abbreviations: C, Celsius; N, number of participants; PCR, polymerase chain reaction; PM, post meridiem; * *p* < 0.05.

## Data Availability

The datasets generated during the current research are not publicly available since patients could be identified by knowing the date of attendance at the health service but are available from the corresponding author upon reasonable request.

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
