# Peer review of "A Preliminary Study on Acute Otitis Media in Spanish Children with Late Dinner Habits"

_ijerph, 2022, doi:10.3390/ijerph191710721_

Round 1
Reviewer 1 Report (Previous Reviewer 2)
Review #2
By reducing this report to a Preliminary Report, the authors have lowered the reader’s expectations of the presentation which is to be commended. A number of formatting issues remain in the tables. It is disappointing to me that a number of pressing suggestions that I made in the previous review were not taken into account. But now as a Preliminary Report, much of these can be overlooked.
A few very minor suggestions are made:
Line 141: “3. Results” is not bolded.
The formatting of Table 2 needs help. The rows are not even, especially for Age, Bedrooms, and Children under 5y. In that last section the two columns 1-4 and 0-3 are still confusing. Is the first row one child or zero children? Please remove either 1-4 or 0-3. What do the asterisks represent? This is not clear and not described in the footnotes.
In Table 3, please remove 1, 2, and 3 from the Disease severity section.
Author Response
Reviewer 1
Thank you so much for taking the time to post a new and positive review of our study
Please see below, for a point-by-point response to your comments and concerns.
Line 141: “3. Results” is not bolded.
ANSWER: Done.
The formatting of Table 2 needs help. The rows are not even, especially for Age, Bedrooms, and Children under 5y. In that last section the two columns 1-4 and 0-3 are still confusing. Is the first row one child or zero children? Please remove either 1-4 or 0-3. What do the asterisks represent? This is not clear and not described in the footnotes.
ANSWER: Thank you for drawing our attention to this issue. Column 1-2-3-4 is a mistake, we have removed it. Column 0-1-2-3 means 0, 1, 2, or 3 children under years in the household, respectively. We have written in the footnotes what the asterisk represent.
In Table 3, please remove 1, 2, and 3 from the Disease severity section.
ANSWER: Done.
Answers Submission Date
18 August, 2022
Reviewer 2 Report (New Reviewer)
The subject is interesting and the introduction contains relevant information and plausible mechanisms. The main finding of the paper is that OM is commoner in later eaters but there has been no effort to correct for potential confounders, and this should be able to be done, given the number of patients in the study. In table 2 the variables of age, health care status , and number of children in the household are all seen to be significantly different between the two groups. They are recognised by the authors and should be corrected for in the analysis.
The material and methods section could be written in a clearer fashion – for instance in lines 117 to 119 “ Confounding Factors” and “ classification of late and early dinner eaters” should be incorporated into sentences.
Line 141 “3.Results” should be given bold status as a major subheading.
In table 2 the age results don’t appear to be in median / IQR format and line 143 presumably should read “ median age”.
Author Response
Reviewer 2
We are very grateful to Reviewer 2 for this new provision of feedback on our manuscript.
Please see below, for a point-by-point response to your comments and concerns.
The material and methods section could be written in a clearer fashion – for instance in lines 117 to 119 “ Confounding Factors” and “ classification of late and early dinner eaters” should be incorporated into sentences.
ANSWER: We thank very much this comment. Accordingly, we have modified this paragraph
Line 141 “3.Results” should be given bold status as a major subheading.
ANSWER: Done.
In table 2 the age results don’t appear to be in median / IQR format and line 143 presumably should read “ median age”.
ANSWER: Children 8 days to 17 years old were included in the study. There is a wide interquartile range. As explained in the footnote of table 2, age results are described as median / IQR measurements of dispersion.
Line 143 reads: “Preregistration: Aspredicted Trials Registry number of COVID IB4221/20PI is”
Answers Submission Date
18 August, 2022
Round 2
Reviewer 2 Report (New Reviewer)
Thank you for making the changes/explanations
This manuscript is a resubmission of an earlier submission. The following is a list of the peer review reports and author responses from that submission.
Round 1
Reviewer 1 Report
in my opinion, the subject discussed by the authors is very important and interesting, however, the collected data is insufficient and should not be the basis for verifying the research problem presented in this work and for formulating such conclusions. When reading this work, one can get the impression that the results presented in it were obtained by accident (by the way of correlating all the data obtained from interviews related to COVID). Here are some arguments:
1) lack of homogeneity of the studied group.
2) incorrect and incomplete nutrition methodology.
3) the study has many limiting factors.
In my opinion, the article is well written, however, using these data it is impossible to pose and verify the research problem posed in the paper. The study requires missing data to be able to answer the researchers' problem presented in the title.
Reviewer 2 Report
The authors perform a secondary data analysis to assess the possible role of late dinner habits in the onset of acute otitis media (AOM). To their credit, it is a creative use of a dataset not intended for this purpose and they have a preliminary finding that deserves follow up. Unfortunately, the p-value is barely significant and has several methodological problems explained below. The result is a barely interesting paper statistically speaking, that needs to be modified before serious consideration.
Abstract
Adding the age of the children would be informative.
Introduction
The authors cite reference 2, a PLoS study of Danish blood donors in support of their assertion that “the timing of caloric intake plays an important role in the long-term process that leads to…infectious diseases.” A careful reading of the PLoS study does not support this statement. Please remove or find a better citation.
Likewise reference 3, an editorial introducing a themed edition of a journal on the circadian rhythm of the gut microbiome, makes no reference to any article that discusses late dinners. This sentence is an interesting conjecture that should be modified by a few “maybe” and “possible” injections.
The same is again true for the next sentence supported by reference 4. This is a fine conjecture that this very manuscript will investigate but should be understood by the reader as an as-of-yet not proven link between late dinners and immune function or chronobiologic mechanisms.
Methods
The first paragraph under Data Collection describes the “common pitfall of emergency room studies”. The statement is unusual but appreciated and helps explain the preliminary nature of the findings. There are definitely some weaknesses to this approach, but the authors are to be commended for taking available data and making the most use of it.
Line 111: Note the formatting issue here.
Line 113: Is “Confounding Factors” a heading? This is not clear.
Lines 115-116: Is “Classification of Late and Early Dinner Eaters” a heading? This is confusing.
Line 124: The real issue with the sample size is not the COVID incidence but the incidence of AOM in this secondary data analysis. The sample size was set by the primary study. The authors should indicate here that the sample size for their hypothesis about AOM was out of their control.
The Abstract indicates that other infectious diseases were not associated with eating patterns. From the Results it appears apart from COVID, no specific infectious disease was diagnosed though several symptoms were questioned. Were there other diseases not mentioned in the Results?
This is admittedly a preliminary study. The key result is from a single chi square result of dinnertime groups vs AOM diagnosis with only 33 cases. We know that age and breast feeding are strong determinants of AOM and these were measured in the study. Also assessed was health coverage, an important determinant of quality of care. Could the authors do a simple logistic regression with these three covariates?
Results
AOM is generally a disease of infants, 6-12 months according to the authors. Yet this cohort contains children up to 17 years old. This seems unnecessary and confusing. The median age is about 4.5 years and the interquartile range is 95.5 which suggests a limited number of older adolescents are in the study. It would greatly improve the study to remove these few older subjects who are very unlikely to get AOM and are badly skewing the statistics. The authors might limit the study to children less than 5 years old. I doubt any cases would be lost.
We note that the median age is about 4.5 years and that breastfeeding is at 68%. This is puzzling and needs explaining. If every single child younger than the median is breastfeeding (that would be 50% of the population) then that leaves 18% of the population over 4.5 years still breastfeeding. This is a rather unusual practice. Am I missing something here?
A distinction was not made between exclusive and continued breastfeeding. This relates to mealtime. Is mealtime for a breastfed child the same as mealtime for the rest of the family? Clearly, this may depend in part on how exclusive the breastfeeding is. While this was clearly not a question on the questionnaire, regrettably, it may be indicated as a weakness of the study in which 2/3 of the children were breastfed to some extent.
The authors have explained the average mealtime of their subjects. It would also be interesting to know the average age of AOM diagnosis.
Table 2.
The Children under 5 y is confusing. Is it 1-4 or 0-3? It appears that 206 households have no kids under 5. Could these be removed from the study?
BMI is improperly expressed. Since children grow so fast, BMI in children is expressed as a percentile for their age. This would be especially true for the current cohort that includes infants and 17-year-olds.
Table 3.
BMI is repeated from Table 2. Not necessary
Summary
I have three strong recommendations.
1) Reduce the size of this study to only those children likely to get AOM, probably under age 5.
2) Perform a simple multivariate logistic analysis including key known covariates of mealtime or AOM, including at least age and breastfeeding.
3) In the Discussion, explain the contribution and complication of breastfeeding as explained above.